# Efficient Multi-Change Point Analysis to Decode Economic Crisis Information from the S&P500 Mean Market Correlation

**DOI:** 10.3390/e25091265

**Published:** 2023-08-26

**Authors:** Martin Heßler, Tobias Wand, Oliver Kamps

**Affiliations:** 1Institute for Theoretical Physics, University of Münster, Wilhelm-Klemm-Straße 9, 48149 Münster, Germany; t_wand01@uni-muenster.de; 2Center for Nonlinear Science, University of Münster, Corrensstraße 2, 48149 Münster, Germany; okamp@uni-muenster.de

**Keywords:** Bayesian multi-change point analysis, linear trend segment fit, computationally efficient open-source python implementation, *S&P500*, mean market correlation, market mode, market factor, economic crises, econophysics

## Abstract

Identifying macroeconomic events that are responsible for dramatic changes of economy is of particular relevance to understanding the overall economic dynamics. We introduce an open-source available efficient Python implementation of a Bayesian multi-trend change point analysis, which solves significant memory and computing time limitations to extract crisis information from a correlation metric. Therefore, we focus on the recently investigated *S&P500* mean market correlation in a period of roughly 20 years that includes the dot-com bubble, the global financial crisis, and the Euro crisis. The analysis is performed two-fold: first, in retrospect on the whole dataset and second, in an online adaptive manner in pre-crisis segments. The online sensitivity horizon is roughly determined to be 80 up to 100 trading days after a crisis onset. A detailed comparison to global economic events supports the interpretation of the mean market correlation as an informative macroeconomic measure by a rather good agreement of change point distributions and major crisis events. Furthermore, the results hint at the importance of the U.S. housing bubble as a trigger of the global financial crisis, provide new evidence for the general reasoning of locally (meta)stable economic states, and could work as a comparative impact rating of specific economic events.

## 1. Introduction

Previous work [1,2,3] has identified the mean correlation of the *S&P500*’s stock time series to capture essential features of the global market movement. These studies uncover a strong agreement between the mean correlations, the maximum eigenvalue of the correlation matrices, and the first principal components of the stock correlations. Starting from random matrix theory, Laloux et al. [2] and Plerou et al. [3] studied correlation matrices of different stock price time series in 1996. They identified the matrices’ greatest eigenvalue to lie far outside the predicted range of purely random matrices. In consequence, it encodes information about collective market behaviour, whereas most of the smaller eigenvalues fall into the expected range for random matrices. The greatest eigenvalue is called the market factor [4] or market mode [5] to account for its peculiar importance. It is mostly equivalent to the mean market correlation [1,5]. Furthermore, Kapwień and Drożdż [4] complement previous results—they find a strongly coupled market core with only weakly linked periphery by investigating the market mode’s eigenvector scaling dependent on the time series length.

Therefore, the mean market correlation compresses lots of information about the market dynamics into a low-dimensional observable space. This has led to extensive research on extracting economic states [1,6,7], identifying crisis precursors, classifying endogenous and exogenous crises [8] and deriving new insights into economic dynamics [9,10].

Following these results, the one dimensional time series should contain valuable information about economic dynamics, such as, e.g., bubbles and economic crises. But if the information is stored in the condensed mean correlations, we should consequently be able to extract that information from the strongly fluctuating time series. A demonstration of that would, first, lead to a better understanding of how the information is encoded in the mean correlation and, second, consolidate—independently from previous works—the significant role of the mean correlation as a global market observable. In this article, we use a Bayesian change point (CP) analysis to identify when the trend of the mean market correlation changes. These changes show a rather good correspondence to the major economic crisis events of the observed time period, i.e., the dot-com bubble, the financial and the Euro crises. A related approach of a CP analysis is used in a publication of Dehning et al. [11] to quantify the efficiency of government decisions during the COVID-19 pandemic. Nevertheless, there are significant differences in our research task: first, the authors of [11] have access to well-established pandemic models—e.g., the susceptible-infected-recovered (SIR) model—whereas we have to assume linear segments with different slopes in the absence of better models for the mean correlation dynamics. Second, Dehning et al. can determine a certain range for CP positions based on known government actions and incorporate gradual changes of parameters directly in the model equations of the Monte Carlo runs. In contrast, we determine the CP probability distributions with respect to all possible CP configurations based on the given data without employing Monte Carlo methods.

We demonstrate that the approach lends itself for extracting information about economic state history, might help to separate endogenous from exogenous crises and implies that it might be possible to derive precursors for endogenous events from the mean correlation. Of course, the approach applied in this article cannot be interpreted in terms of precursor information, since it is applied offline and in retrospect of the *whole* mean correlation time series, i.e., it contains data from before and after the major economic events. This also holds if we change into an online perspective in which we successively add future data and perform the CP analysis on each updated time series. The CP will only manifest itself if we include a minimum amount of data that encodes the uprising trend change. However, the online sensitivity horizon of the method to detect trend changes in the investigated time series lies roughly between 80 and 100 trading days, which might provide a chance to identify current events for experts to focus on with respect to possible new and locally quasi-stationary economic states. Depending on the time horizon of macroeconomic research it might even provide some evidence for ongoing changes of the market dynamics.

Nonetheless, the separation of the investigated time period into segments of almost constant linear trends offers a rather straight-forward interpretation in terms of temporarily stable market periods. Even if it is not possible to interpret these states as abstract conceptional entities—as it would be possible, e.g., by the clustering approach that is described in Stepanov et al. [1]—the method allows for a direct evaluation of the data and a closer connection to real events. It is also robust with respect to intrinsic parameter choices, which is not guaranteed for clustering algorithms and the pre-assumed number of clusters.

The remainder of this article is structured as follows: Section 2 explains the data gathering and preprocessing in Section 2.1 and the Bayesian methodology used for the CP detection in Section 2.2. Section 3 shows the ex post identified CPs in Section 3.1 and an online analysis in Section 3.2. The results are compared in detail to a timeline of global economic events of the period. Finally, we discuss and compare our results to the work of Stepanov et al. [1] in Section 4.

## 2. Data and Methods

The preprocessing of the data that we analyse is shortly summarised in Section 2.1. More details about the CP analysis method can be found in Section 2.2.

### 2.1. Data Preparation

The procedure chosen to calculate the mean market correlation of the *S&P500* stock index is chosen analogously to the articles [1,12]. Confined to the considered time period between 1 January 1992 and 28 December 2012, we filter for companies that are included in the *S&P500* index for at least 99.5% of the time and interpolate occasionally missing data. The data are used to compute pairwise correlations $C_{i,j}$ of the locally normalised returns of the remaining 249 daily-resolved stock price time series for 5291 trading days. The correlations are computed on overlapping rolling data windows of size τ=42. Shown by the principle component analysis in [1], the mean correlation,
(1)$$\bar{C}=\frac{1}{N}\sum_{i,j}C_{i,j},$$
is in high agreement with the first principle component. In addition, it describes the greatest part of the variability in the data. Thus, we focus on the mean market correlation $\bar{C}\left(t\right)$, in which the mean correlation of each window is identified with its 21st time stamp. The corresponding time series is shown in blue in Figure 1. More details about the preprocessing procedure can be found in Appendix A.

### 2.2. Change Point Identification

Since the mean market correlation time series $\bar{C}\left(t\right)$ should contain valuable information about changing market characteristics, we investigate its changing trends over a time period of stable and crises states in detail. Therefore, we use a tool that considers the CPs of linear trends over time *t*. The open-source Python package *antiCPy* [13,14,15] implements the Bayesian analysis of changing linear trends, introduced by von der Linden et al. [16,17]. By assuming *m* CPs in the trend, the time series data $\bar{C}\left(t\right)$ are modelled by linear segments $\phi_{K}$ of different slopes, formally given by
(2)$$\phi_{K}\left(t|\underline{E},{\underline{\bar{C}}}_{\mathrm{ord}},I\right)={\bar{C}}_{K}\frac{t_{K+1}-t}{t_{K+1}-t_{K}}+{\bar{C}}_{K+1}\frac{t-t_{K}}{t_{K+1}-t_{K}},$$
with $t_{K}⩽t⩽t_{K+1}$ and the CP vector $\underline{E}$ with the entries $E_{K}=t_{K+1}$ for K=1,2,...,N−2, whereby *N* denotes the size of the correlation time series $\bar{C}\left(t\right)$ and the vector ${\underline{\bar{C}}}_{\mathrm{ord}}$ contains the design ordinates ${\bar{C}}_{K}$ of the assumed CP positions, i.e., the positions where consecutive segments are connected with each other. Note that the first and last allowed position of a CP is restricted to the second and penultimate time stamp *t*. For each possible configuration $\underline{E}$ of CPs the corresponding posterior probability $p(\underline{E}|\bar{C}\left(t\right),\underline{t},I)$ is calculated via Bayes’ theorem,
(3)$$p\left(\underline{E}|\bar{C}\left(t\right),\underline{t},I\right)=\frac{p\left(\underline{E}|\underline{t},I\right)\;\cdot \;p\left(\bar{C}\left(t\right)|\underline{t},\underline{E},I\right)}{p(\bar{C}\left(t\right)|\underline{t},I)},$$
with the underlying time steps $\underline{t}$ and the background information I. Following this reasoning, we can derive the most probable CP configurations quantitatively if we perform the calculations for all feasible CP combinations, i.e., a number of
(4)Zm=N−2m=(N−2)!m!((N−2)−m)!
CP configurations and select the most probable ones. Furthermore, the complete probability density function (PDF) of CP configurations enables us to calculate the most probable linear segment fit and confidence bands (CBs) for each time step by calculating the averaged sum of the linear segment fits at that time weighted with their respective probabilities. Doing so, even future values and their CBs can be extrapolated based on the current data information in a formally consistent manner. In addition, the marginal CP PDFs can be computed by averaging the joint probabilities at each time stamp *t* with respect to its ordinal position in each configuration.

However, performing these calculations proves to be almost prohibitive (cf. also [17]) if we recall the combinatorial number of CP configurations, e.g., for the whole time series $\bar{C}\left(t\right)$ of 5269 data points for only three CPs, i.e.,
(5)Z2=52673≈2.4·1010
combinations. One idea to solve this issue for long time series with a moderate number of CPs is presented by von der Linden et al. [17]. The approach uses a Monte Carlo sum over a limited number of random CP configurations to estimate the linear segments. However, with an increasing number of CPs, even a Monte Carlo sum approach is not feasible anymore, since the extremely high-dimensional CP configuration space is sampled too sparsely to derive reliable fit results. Furthermore, the time to compute the linear segment fits serially explodes up to several decades or hundreds of years. For this reason, first we decide to thin the mean market correlation data $\bar{C}\left(t\right)$ by storing only every 40th data point and performing the calculations on this dataset of 132 points. The resulting dataset preserves the trend characteristics of the data very well (cf. Figure 1). Nevertheless, a naive serial implementation of the CP analysis would fail with regards to the computation time and—maybe even worse—collapses due to memory limitations which do not allow for storing all possible CP configurations at the same time. Imagine, e.g., the situation of the thinned time series under the assumption of five CPs—we easily run into problems if we want to store a matrix of dimensions (286,243,776 × 7) with double precision. To avoid such issues, we take advantage of the efficient implementation of the CP analysis in the *antiCPy* package. It involves parallelisation and an algorithm that constructs only a suitable subset of CP configurations and keeps these subsets safely organised for each parallel worker. This avoids any redundancy that would compromise the results. More details of the practical implementation will be prepared in a follow-up manuscript.

## 3. Results

In Section 3.1, we apply the CP analysis to the whole mean market correlation dataset $\bar{C}\left(t\right)$ under the assumption of two up to four CPs. The results are double-checked by analyses with one and five CPs as documented in Appendix B. In addition, we perform an online adaptive CP analysis in the three pre-crisis segments in Section 3.2 to complete the analysis.

### 3.1. Change Point Analysis

As already mentioned above, the one-dimensional mean market correlation $\bar{C}\left(t\right)$ is expected to contain valuable information about the global market dynamics including major crisis events. However, extracting this information in a reliable manner is an ambitious task. Based on our assumption that the linear trends of the market correlation encode some information about changing dynamics due to crisis events, we perform a detailed CP analysis as described in Section 2.2. We compute the PDFs of the CPs assuming one up to five trend-CPs hidden in the data to confirm that our results are stable against these a priori assumptions. Even the results based on only one assumed CP, which obviously cannot reflect the complicated trend structure in the data, are roughly consistent with the more elaborate fits. The same holds true for models with more than four CPs. The interested reader can take a look at the results for one and five CPs in Appendix B, Figure A1.

The most suitable assumptions should contain at least three CPs, since the investigated period from 31 January 1992 until 28 December 2012 contains three important major economic events: the dot-com bubble, the financial and Euro crises. Therefore, in Figure 2 of the main article, we show our results from two up to four CPs in increasing order and compare them to these three major economic events. Unfortunately, the continuously progressing dynamics of the economy can make it rather difficult to determine the exact dates of crisis onset and decline.

In Table 1, we provide a timeline of selected significant events of the three major economic bubbles and crises and compare them to the time stamps at which maxima occur in the CP PDFs of Figure 2. Some maxima, especially of the higher order CP PDFs, are only visible by appropriate up-scaling. For example, the high peak of the first CP PDF in Figure 2c is accompanied by a very small one in Figure 2a. In Figure 2b it is not found.

#### 3.1.1. Dot-Com Bubble

First, let us consider the dot-com bubble—we highlight in grey the approximate time interval in which the *NASDAQ Composite Index* starts to grow relatively fast from 1 January 1999 up to 1 October 2002, the day at which the *NASDAQ Composite Index* fell to roughly its lowest value after the bubble’s all-time high, marked by the black dotted vertical line on 3 October 2000. The trough of 1 October 2002 is accompanied by a more general stock market downturn in 2002 which started around the 11 September 2001 terrorist attacks [18]. The most interesting observation for the dot-com bubble is that around its onset and decline the marginal CP PDFs of the second (cf. Figure 2a), second and third (b) or third and fourth (c) CP are pronounced as visible in the insets. For orientation in the insets, the beginning and end of the dot-com interval are indicated by black and grey solid vertical lines, respectively. In Figure 2a,c the CP PDFs of the first (a) and first and second CP (c), respectively, also reach local maxima shortly after the bubble’s all-time high. These are plausible results, since the dot-com bubble was of speculative nature, which should become visible in the studied correlations of the *S&P500* stock index returns without notable delay. For completeness, we mention the 14 May 1999 as an important economic date independent from the dot-com bubble. On this day a sharp rise in consumer prices in the U.S. led to a drop in the stock market [19]. Even if the width of the CP PDFs and their pronounced profile at the end of the bubble tend to support mostly a connection with the continuous strong rise of the *NASDAQ Composite Index*, of course our approach cannot decide definitely whether events like 14 May 1999 or the aftermath of the Asian financial crisis [20] around 1997–1998 or Russian financial crisis [21] around 1998–1999 might also influence the CP PDF profiles. Keeping that in mind, the correspondence of CP positions and the onset/decline of the dot-com bubble nevertheless strengthens the hypothesis that the mean market correlations trends encode distinct market periods.

#### 3.1.2. Financial Crisis

Second, we consider the financial crisis. While the dot-com bubble was mostly created by speculative investment decisions, less speculatively induced economic crises like the global financial crisis include a complex cascade of sub-crisis events and cannot be simplified in that manner. Therefore, we proceed with indicating their approximate beginning up to their culmination events. The very pronounced peaks of the first (a), the first and second (b) and the first and third (c) CP PDFs are located rather exactly in the time interval from 31 October 2006 up to 3 January 2007 which coincides with the burst of the U.S. housing bubble, the trigger event for the global financial crisis of the later years [22]. Nevertheless, the commonly accepted definition of the beginning of the financial crisis is on 9 August 2007 when the interest rates for inter-bank financial loans rose sharply [23,24]. This emphasises the problem of determining exact dates for crisis events. Our results suggest that the U.S. housing bubble is clearly notable as a change in the mean market correlation, whereas the change in interest rates for inter-bank financial loans is not detected by our CP analysis. This is a rather expected result, since we consider stock return correlations—the correlations probably capture financial events, like strong price changes of stock bubbles, rather accurately, but their response to real-world events (like the changing interest rates) might be delayed.

However, to demonstrate the precise coincidence, we mark the burst of the U.S. housing bubble around 1 January 2007 as the beginning of the grey shaded area for the global financial crisis and end with the Lehman Brothers’ insolvency on 15 September 2008 [25]. Also for the Lehman Brothers’ bankruptcy, at least the first and second CP PDFs exhibit local maxima regardless of assumed number of CPs. Note that, similar to the dot-com bubble’s key dates, there is almost no delay between the event and the detected trend change, since the Lehman Brothers’ bankruptcy had an almost immediate effect on the world economy. Only two weeks later, on 29 September 2008, the U.S. stock market collapsed [26].

#### 3.1.3. Euro Crisis

Third, we discuss the Euro crisis as the direct consequence of the global financial crisis. We indicate its earliest start when the first ten European banks asked for a bailout up to January 2009, represented in the figure by the 1 January 2009, and end with the culmination of the crisis when Greece’s sovereign debt rating was downgraded on 27 April 2010 by *Standard & Poor’s* [27]. The temporal proximity and strong causal connection of the global financial crisis and the Euro crisis are captured by the width of the local maxima of the second (a), third (b) and first, second, and fourth (c) CP PDFs in the passage from one crisis to the other. Also, the commonly mentioned culmination event of the downgrading of Greece’s sovereign debt rating is mirrored by the CP analysis results. Even if the downgrading alone might have only a moderate influence on the mean market correlation trends in terms of symbolic manifestation of the crisis, the indicated culmination event is mirrored by small peaks in the second and first CP PDF in (a) and (b), respectively, and in the well-pronounced peaks of the first and second CP PDFs in (c).

Apart from these considerations, the CP PDFs in the cases (a–c) show pronounced maxima in the second half of 2010 up to December 2011. It seems to be impossible to deduce origin events for each individual CP PDF peak, since the time period is full of economic (and political) events including a series of sovereign debt downgradings of European countries, bailout plans, the Arab Spring, and more, summarised in Table 1. But in spite of that, it seems to be rather probable that the trend changes detected by the above mentioned CP PDFs are correlated with the turbulent economic situations at that time.

**Table 1 entropy-25-01265-t001:** Overview about economic events. The table is not claimed to be complete, but serves as a chronological orientation. The CP PDF peaks of Figure 2 are printed for easy comparison, but should be interpreted carefully, since the analysis does not allow for causal inference.

Event Date	Event Description	Source	CP PDF Peak Dates		
			Figure 2 **a**	Figure 2 **b**	Figure 2 **c**
1994	Over the year, bond prices fell continuously in consequence of partially unexpected and repeated raise of federal funds rates by the *FED*. In consequence bonds lost about USD 1.5 trillion in market value globally.	[28]	2 February 1995 to 26 July 1995; 31 March 1995 to 21 September 1995	2 February 1995; 31 March 1995 to 21 September 1995	26 July 1995 to 16 November 1995; 2 February 1995; 26 July 1995 to 16 November 1995
December 1994	Due to devaluation of the peso by the Mexican government and thus anticipating further devaluations, investors rapidly withdrew capital from Mexican investments. In January 1995 the U.S. government coordinates a USD 40 billion bailout.	[29]	2 February 1995 to 26 July 1995; 31 March 1995 to 21 September 1995	2 February 1995; 31 March 1995 to 21 September 1995	26 July 1995 to 16 November 1995; 2 February 1995; 26 July 1995 to 16 November 1995
1997–1998	Asian financial crisis in East and Southeast Asia. Multiple origins are discussed. However, long-lasting global contagion stayed away and the markets recovered fast in 1998–1999.	[20]	14 May 1999	10 October 1997; 18 March 1999	10 October 1997; 18 March 1999; 14 May 1999
1998–1999	Russian financial crisis in which the ruble was devalued. As a result, many neighbouring countries experienced also severe crises.	[21]	14 May 1999	18 March 1999 to 13 July 1999	18 March 1999; 14 May 1999
1 January 1999	The modest onset of the dot-com bubble 1998 turns into a rapid rally of the *NASDAQ Composite Index*. In the literature the long pre-bubble period begins commonly in 1995.	[30,31]	14 May 1999	18 March 1999 to 12 July 1999	18 March 1999
14 May 1999	Stock market drop due to sharp rise of consumer prices to 1.75% in the U.S.	[19]	14 May 1999	14 May 1999 to 12 July 1999	14 May 1999
10 March 2000	Based on the *NASDAQ Composite Index* the Dot-com bubble reaches its all-time high.	[30,31]	16 October 2000	22 June 2000; 9 April 2001	kink at 22 June 2000; 9 April 2001
11 September 2001	In course of a terror attack three airplanes crash into the twin towers of the U.S. World Trade Center and the Pentagon coordinated by the militant Islamist extremist network al-Qaeda.	[18]	24 May to 23 July 2002	4 October 2001; 24 May 2002; 23 July 2002	4 October 2001; 30 January 2002; 24 May 2002; 23 July 2002; 18 September 2002
1 October 2002	Based on the *NASDAQ Composite Index* the Dot-com bubble reaches a new trough after the bubble burst in course of a more general stock downturn since 11 September 2011.	[30,31]	28 July 2005 to 18 November 2005	28 October 2003 to 24 December 2003	28 October 2003 to 24 December 2003
2002/2003	The Venezuelan General Strike massively hinders oil exports among others to the USA.	[32,33]	28 July 2005 to 18 November 2005	28 October 2003 to 24 December 2003	28 October 2003 to 24 December 2003
2003	Approximate beginning of the boom-phase of the U.S. housing bubble.	[34]	28 July 2005 to 18 November 2005	28 October 2003 to 24 December 2003	28 October 2003 to 24 December 2003
2003–2016	Begin of rising oil prices from under USD 25/bbl to above USD 30/bbl. The peak of USD 147.30/bbl is reached in July 2008, before they return temporarily to USD 35/bbl in 2009.	[33,35]	28 July 2005 to 18 November 2005	28 October 2003 to 24 December 2003	28 October 2003 to 24 December 2003
Q4 2006 to Q1 2007	Period of most intense burst of the U.S. housing bubble with new lowest price in 2011.	[22]	2 November 2006 to 3 January 2007	31 October 2006; 2 January 2007	2 November 2006 to 3 January 2007
9 August 2007	Commonly listed onset of the global financial crisis. On this day the interest rates for inter-bank financial loans rose sharply.	[23,24]	22 August 2007	22 August 2007	22 August 2007
2008	Year of the financial crisis and great recession. The year is characterised by manifold bailouts for banks and stock market crashes.	[36]	overall high CP PDFs in 2008	overall high CP PDFs in 2008	overall high CP PDFs in 2008
January to March 2008	The housing prices continue to collapse. The government passes a tax rebate bill on 13 February 2008 and the *FED* starts bailout programs in the beginning of March.	[37,38]	4 August 2008	4 August 2008	4 August 2008
30 July 2008	U.S. governments passes bailout laws for Fanny Mae and Freddie Mac.	[39]	Kink at 4 August 2008	4 August 2008	4 August 2008
7 September 2008	U.S. government’s take over of Fanny Mae Association and Freddie Mac Corporation.	[40]	1 October 2008 to 26 November 2008	1 October 2008	1 October 2008 to 26 November 2008
15 September 2008	Lehman Brothers bankruptcy.	[25]	1 October 2008	1 October 2008	1 October 2008
29 September 2008	Stock market collapsed when the bailout bill was rejected by the U.S. House of Representatives.	[26]	1 October 2008	30 September 2008	1 October 2008
24 October 2008	Many world stock indices lost around 10%.	[41]	26 November 2008	26 November 2008	26 November 2008
1 December 2008	When the National Bureau of Economic Research officially declared that the U.S. was in a recession since December 2007, the *S&P500* lost 8.93% and the financial stocks of the index even 17% based on these news.	[42]	high CP PDF after 26 November 2008	high CP PDF after 26 November 2008	high CP PDF after 26 November 2008
January 2009	The first ten European banks ask for bailout programs.	[43]	high CP PDF after 26 November 2008	high CP PDF after 26 November 2008	high CP PDF after 26 November 2008
Early 2010 to mid 2012	The Arab Spring. Great anti-government protests in large parts of the Middle East and North Africa. The oil prices rise above USD 100/bbl.	[35,44,45]	5 May 2010 to 1 July 2010	5 May 2010 to 1 July 2010	5 May 2010 to 1 July 2010
27 April 2010	Greece’s sovereign debt rating was downgraded by *Standard& Poor’s*.	[27]	5 May 2010 to 1 July 2010	5 May 2010 to 1 July 2010	5 May 2010 to 1 July 2010
6 May 2010	So-called flash crash led to a 9% drop in the *Dow Jones Index* caused due to high frequency trading.	[46]	5 May 2010 to 1 July 2010	5 May 2010 to 1 July 2010	5 May 2010 to 1 July 2010
8 May 2010	Passing drastic bailout plans for Greece in Brussels. A EUR 110 billion package is approved.	[47,48]	5 May 2010 to 1 July 2010	5 May 2010 to 1 July 2010; 28 August 2010	5 May 2010 to 1 July 2010; 1 July to 28 August 2010
17 May 2010	The Euro currency falls into first four-years low.	[49]	5 May 2010 to 1 July 2010	5 May 2010 to 1 July 2010; 28 August 2010	5 May 2010 to 1 July 2010; 1 July to 28 August 2010
Q1/2 2010	Several downgradings of debt ratings/bonds of European countries, i.e., Portugal, Spain, Greece.	[50]	5 May 2010 to 1 July 2010	5 May 2010 to 1 July 2010; 28 August 2010	5 May 2010 to 1 July 2010; 1 July to 28 August 2010 3 September 2010
4 June 2010	The Euro currency falls into a second four-year low. Major American markets fall more than 3%.	[51]	5 May 2010 to 1 July 2010	5 May 2010 to 1 July 2010; 28 August 2010	5 May 2010 to 1 July 2010; 1 July to 28 August 2010
November 2010	Bailout request by Ireland. At the end of November Ireland receives EUR 85 billion.	[52,53]	15 April 2011	15 April 2011	15 April 2011
7 April 2011	In the evening of 6 April 2011 Portugal’s government announces that it is the third after Greece’s and Ireland’s governments that will ask for a bailout package. On 17 May 2011 a bailout package of around EUR 78 billion is formally adopted.	[54]	15 April 2011 to 14 June 2011	15 April 2011 to 14 June 2011	15 May 2011 to 14 June 2011
13 June 2011	Greece’s credit rankings become worst in the world.	[55]	13 June 2011	13 June 2011	13 June 2011
August 2011	Under the *Securities Markets Programme* the ECB restarts to purchase a significant amount of eurozone sovereign bonds. Spanish and Italian yields breach 6%.	[56,57]	6 October 2011	9/10 August 2011	10 August 2011

#### 3.1.4. Remarks and Intermediate Conclusion

Since the CP PDFs are computed over the thinned grid of time steps only every 40th trading day is sampled. For the definition of the maxima we examine these really calculated support points. In cases of modes of a certain width we yield intervals of higher CP PDF. Of course, the listed events are not complete and its content should not be considered as causal deduction between the listed real world events and the CP PDF peaks. It should rather illustrate that the uncovered times of high trend change PDFs in fact correlate with highly turbulent times in the economy. The maxima of the CP PDFs are clustered around events that are connected with the dot-com bubble, the global financial crisis and the Euro sovereign debt crisis.

All in all, the results support the hypothesis that the trend in the mean market correlation encodes valuable information about major economic events that can be uncovered by applying a Bayesian trend CP analysis. As the similarities of the time stamps for CP PDF peaks in Table 1 illustrate, the results are comparable to a high extent, regardless of how many CPs we assume. For an extended robustness check compare the results in Figure A1 under the assumption of one and five CPs in Appendix B. It might be unsurprising that sharp drops and bursting bubbles like the dot-com bubble or the U.S. housing bubble are often well-captured by our approach, since we rely on correlations of stock returns which show these changes almost immediately. Nevertheless, considering the mean market correlation with a CP analysis enables us to eliminate trend changes that manifest only in a small subset of stock prices and thus, acts like a filter to focus only on economic events of global importance. Furthermore, in some cases the ex post derived CP PDF peaks could help to identify important economic events or even rate the impact of specific time periods or isolated economic events. Moreover, the general ansatz can be adapted to other economic observables or subsets of stock prices to filter for economic events that are of special interest for the considered subset (e.g., national companies, companies of certain sectors or of certain size etc.). However, we emphasise that the analysis is performed on the whole available data which makes it an ex post analysis that cannot be interpreted in terms of an ex ante economic risk predictor.

We conclude the CP analysis discussion with some remarks on the consistency of the results. The blue mean market correlation $\bar{C}\left(t\right)$ in Figure 2 is well-fitted by the orange segment fits with corresponding 3σ-CBs. The CBs quantify the certainty of the given fit based on the surrounding data information. In that sense, the narrow shape of the CBs suggests that the fit performs well for the underlying mean market correlation data $\bar{C}\left(t\right)$. A comparison of the very similarly shaped fits (a–c) also confirms the robustness of the results. Finally, we studied the individual most probable CP configurations three of which are drawn by green dashed, violet dash-dotted and red dotted vertical lines in Figure 2 as an example. In (a–c) the most probable configurations are clustered around the CP PDF peaks of the global financial crisis and the Euro sovereign debt crisis. The dot-com bubble is included only a bit later in the most probable configurations, but consistent with the less pronounced CP PDFs of that period. However, the first configuration that includes a date between the 1 January 1999 and the 1 October 2002 is identified to be the 264th of 8385 total configurations in (a), the 26th combination of 357,760 in (b) and the 354th of 11,358,880 configurations in (c). This corresponds to the most probable 3.15%, $7.28\;\cdot \;10^{-5}$% and $3.12\;\cdot \;10^{-5}$% configurations in (a), (b) and (c), respectively, and round out our analysis.

### 3.2. On-Line Change Point Evolution in Pre-Crisis-Segments

In the CP analysis of the mean market correlation $\bar{C}\left(t\right)$ from 31 January 1992 until 28 December 2012 in Section 3.1, we take a retrospective point of view to investigate whether the linear trend changes in the mean market correlation $\bar{C}\left(t\right)$ encode information about economic crisis events of global importance. Since our findings in Section 3.1 suggest indeed such a connection, in this subsection we complement the consistency of our results by changing from a retrospective to an online perspective. From the online viewpoint three key questions arise: First, is the time evolution of the CP PDFs $p(E|\bar{C}\left(t\right),\underline{t},I)$ of one CP *E* consistent with our expectations if we apply it to currently available data and update these data whenever new trading days end? Second, how much data of a changing mean market correlation trend, which potentially marks an important economic period, has to be included to render the CP analysis sensitive? And third, do the results change consistently on the stronger timely confined data segments, i.e., do the CP PDFs mirror in a way the relative impact of the events included in the sub-intervals? To answer these questions, we assume one CP and compare the CP analysis results calculated on two slightly differing types of time series segments for the dot-com bubble, the global financial crisis and the Euro crisis. The first type is limited exclusively to pre-crisis data of the mentioned major crisis events and the second type includes a small amount of mean market correlation data beyond the crises’ onsets at $t_{0}$, i.e., it is cut after $t_{1}>t_{0}$. The results are presented in Figure 3. For the pre-dot-com bubble segment as well as for the financial crisis data we used every tenth data point to avoid numerical instabilities due to an overflowing normalisation factor of the CP PDFs. The short segment of the Euro crisis is not thinned at all. Analogously to Section 3.1 we show the accompanying segment fits of the orange and the black CP PDFs in Figure 3 to illustrate the most probable linear segments each of which might resemble specific economic periods.

#### 3.2.1. Dot-Com Bubble

We start considering the pre-dot-com bubble data in Figure 3. The blue segment is cut from the beginning of the $\bar{C}\left(t\right)$ time series on 31 January 1992 up to the approximate onset $t_{0}$ of the rapid increase of the *NASDAQ Composite Index* on 1 January 1998. Therefore, it does not contain any dot-com period data and the CP PDF should accumulate either around the times of economic events of major impact that are already confined in the data segment or, in the absence of such events, be less pronounced and probably located almost randomly due to some noisy outliers in the data (i.e., the probability of finding the mode of the CP PDF exactly in the end period of the data is relatively low). Note that, in any case, we have to expect some mode of the CP PDF $p(E|\bar{C}\left(t\right),\underline{t},I)$ regardless of whether there is a crisis event or not. This traces back to the design of the trend CP analysis in which we compute the CP PDF $p(E|\bar{C}\left(t\right),\underline{t},I)$ conditional on the background information I, i.e., our prior knowledge/assumption that at least one CP is present in the data. Keeping that in mind, the CP PDFs will exhibit a flat profile only in the rare situation of almost no fluctuations of constant trends, i.e., in the limiting case of an almost unperturbed straight line.

However, the CP PDF $p(E|\bar{C}\left(t\right),\underline{t},I)$ indeed accumulates over the year 1995 coinciding with two important economic events: the so-called 1994 bond market crisis [28] with sharply decreasing bond prices over 1994 (and a strong drop in the yield spread of 1-year and 30-year U.S. Treasury Bonds from roughly 2.83% in 25 January 1994 to 0.54% in 13 December 2004 [58], i.e., an indicator for a heavily flattened yield curve and worse economic conditions), and the Mexican peso crisis [29] starting in December 1994 which spread to other emerging markets. Updating the time series segment with new available data and based on the results of Section 3.1, we expect that the CP PDF is shifted consistently to the dot-com bubble onset period in 1999, once data beyond the onset is included. And in fact, the black CP PDF computed on the updated red mean market correlation $\bar{C}\left(t\right)$ including data up to 1 June 1999 is shifted from 1995 to January 1999, corresponding to the dot-com bubble CP results in Section 3.1.

Note that the appearance of the black CP PDF’s peak cannot be interpreted in terms of a precursor of the upcoming bubble for two reasons. First, we need some data points beyond the crisis onset and second, from the online point of view the shifted CP PDF might be in turn corrected by newly updated data points that uncover the detected trend change as a negligible fluctuation in the long run. Nevertheless, once detected, subsequent inclusion of new information can also confirm the shifted CP PDF, being the case for the major events detected in Section 3.1. In that way the online implementation can be very helpful to focus economic research attempts early onto current time periods which in terms of impact supersede the last major event—here, i.e., the bond crisis in 1994 and the Mexican peso crisis in December 1994—and have potentially stronger influence to future economic states. Such a line of argument might be supported to a certain extent by the clearly more pronounced and narrow profile of the black CP PDF in Q1 1999 compared to the wider and less pronounced orange CP PDF of the years 1995, corresponding to a sharper and more probable trend change in 1999 than in 1995.

Additionally, we determined approximately the minimum amount of data $\bar{C}\left(t\right)$ that are needed to shift the global maximum of the CP PDF to a date $t_{\mathrm{shift}}$ with $t_{\mathrm{crisis}}-100<t_{\mathrm{shift}}<t_{\mathrm{crisis}}+100$, whereby in the thinned time series the bubble’s onset date $t_{\mathrm{crisis}}$ is defined as the date that follows directly on the bubble’s onset on 1 January 1999, i.e., the 5 January 1999. In detail this procedure leads to the 30 April 1999 marked by the green dotted vertical line in Figure 3. Thus, to make the CP analysis sensitive to the bubble onset we need approximately raw stock return data up to the 1 June 1999 (i.e., 21 trading days later), since the mean market correlation $\bar{C}\left(t\right)$ is derived from windows of 42 trading days returns and identified with the 21st window date.

The resulting green CP PDF computed on data up to the green dotted vertical line at date $\tilde{t}$ fits into the previous reasoning as it represents a snapshot at date $\tilde{t}$ of the time evolution from the orange CP PDF at $t_{0}$ to the black CP PDF at date $t_{1}$ with $t_{0}<\tilde{t}<t_{1}$. The old global maximum of the orange CP PDF shrank down to the flat double peaked relic of the green PDF. At the same time the new global maximum in January 1999 has already manifested itself. By adding new data the green global maximum will eventually grow to the black CP PDF and the green remnants in 1994/1995 vanish.

#### 3.2.2. Financial Crisis

For the financial crisis, we observe a very similar behaviour. The segments start beyond the dot-com bubble’s trough on 1 October 2002. The pre-crisis segment is truncated with the burst of the U.S. housing bubble as the seed event of the financial crisis on 1 January 2007, whereas the longer red time series includes in-crisis data until 9 August 2007 when the interest rates for inter-bank financial loans rose sharply. Similar to the previously discussed dot-com bubble segment the orange CP PDF might reflect important events in the confined context of the segment. Consistently with the coincidence of the CP PDFs and the U.S. housing bubble’s burst around January 2007, the onset of the bubble’s pronounced rise began in 2003 [34]. Additionally, rising oil prices in 2003 from USD 20/bbl to over USD 30/bbl peaking in 2008 with USD 147/bbl, a phenomenon referred to as the 2000s energy crisis [33,35], may have contributed to the trend change. Also the Venezuelan general strike in 2002/2003 might play a role because it hindered oil exports heavily. In this time the U.S. imports of oil from Venezuela show a pronounced trough [32,59]. The ambiguity and plurality of possible events underlines the obvious limitations of such a CP analysis which is unable to distinguish the cause of the trend change when the events overlap. However, it might help to reduce the number of probable events that could be responsible for new economic states to the area of pronounced CP PDFs.

The green CP PDF is computed with data until 1 March 2007, i.e., the approximate date from which the CP PDF is sensitive to the U.S. housing bubble burst is around 30 March 2007. The strongly accumulated probability mass in the narrow green global maximum is smoothed out a bit to the black CP PDF when the data are updated.

#### 3.2.3. Euro Crisis

In contrast to the former analyses, the pre-Euro crisis segments tell a different story due to the fact that the crisis is embedded in turbulent economic times which make it rather difficult to isolate a precise period of the Euro crisis. In fact the mean market correlation shows gradually increasing linear downwards trends interrupted by less pronounced upwards trends, i.e., an overall downwards trend in the long run. Therefore, the detected mode of the CP PDF is more sensitive to the chosen time series period. This is in good agreement with our CP analysis in retrospect (cf. Section 3.1), where the CP PDF modes in the appropriate period are also widened from 2008 until 2010. Anyway, the applicability of the CP analysis is limited due to the alternating short term trends and the non-isolated nature of the Euro crisis emerging almost directly after the global financial crisis.

In any case, we follow our previous approach for the starting time of the investigated time series segments, i.e., we start with the culmination event of the previous crisis, here the Lehman Brothers’ bankruptcy on 15 September 2008. Depending on the end of the segment the CP PDF is shifted consistently to the right, i.e., the emerging Euro crisis supersedes older events but cannot be fixed exactly. It fluctuates depending on the amount of included in-crisis data which of course add more short term trend information and compromise the assumption of only one CP more and more. Varying the end date from our defined onset of the Euro crisis on 1 January 2009 to 1 June 2009 leads to the CP PDF fluctuating roughly between 1 January and 1 April 2009. However, the onset of the negative long term trend is slightly before 1 January 2009. The corresponding analysis of segments ending on 1 December 2008 and 1 January 2009 is presented in Figure 3 and underlines the missing isolation of the Euro crisis period from the financial crisis and the interconnected Great Recession of 2008 when the U.S. gross domestic product fell 4.3% which was the deepest recession since World War II [60,61] at that time.

In contrast to the former two analysed segments, the orange and black CP PDFs do not differ a lot in shape, suggesting that the related trend changes cannot be classified in terms of lower or higher impact, but seem to be of similar nature constructing the downwards trend in the long run. This seems to be feasible because over 2008 the critical economic conditions led to several giant rescue plans for banks and a cascade of severe stock market crashes. From 6 to 10 October 2008 the *Dow Jones Industrial Average* fell 18.2%, the *S&P500* even over 20% [62]. Some crashes created the need of interim interruption of the stock markets, e.g., in Indonesia (from 8 to 13 October 2008) due to drop of 10% in one day or Iceland (9, 10 and 13 October 2008) [63,64]. On 24 October 2008 again losses of around 10% were realised in most of the indices [41]. This, as well as the IMF prediction of a worldwide recession of −0.3% on 6 November 2008 and accompanying a lowering of target interest rates by the Bank of England and the ECB most probably contribute to the detected CP of the orange PDF [65]. By updating the data up to $t_{1}$, finally the black CP PDF emerges slightly after 1 December 2008 when the National Bureau of Economic Research officially declared that the U.S. was in a recession since December 2007. In consequence of these news the *S&P500* lost 8.93% and the financial stock of the index even 17% [42]. Further hints supporting the hypothesis that the event cascade drives the downwards trend is given by the green interim CP PDF result: As the data are updated, the green interim CP PDF is first widely smeared out between the old orange and the new black PDF peak and becomes a bi-modal CP PDF with the peaks near these two positions. In that way, the time evolution of the CP PDF resembles the cascade of similarly important events that contribute to the decline of the mean market correlation in the long run. In principle, also in the case of further added data the CP PDF travels along the cascade of events via a bi-modal interim PDF from peak to peak in a similar pattern.

The method is sensitive to the short term trend change if data until 23 December 2008, marked by the green vertical dotted line, is included. This corresponds to an earliest CP analysis sensitivity from 26 January 2009 on.

## 4. Discussion and Conclusions

In Section 3.1 and Section 3.2, we present the results of two CP analyses: first, in retrospect over the thinned mean market correlation $\bar{C}\left(t\right)$ time series and second, in an adaptive approach that updates a pre-crisis segment continuously as present trading days end. Based on an extensive literature research we provide a detailed comparison to the overall economic history of roughly 20 years from 31 January 1992 to 28 December 2012 that includes three major economic crises and we check the consistency of our results against a variation of the method’s intrinsic parameter, i.e., the number of expected CPs. Applying our computationally efficient implementation of the CP analysis from one up to five CPs (roughly $3\;\cdot \;10^{8}$ combinations) we find a weak dependence of the main results of the CP distributions on the expected number of CPs from two up to five CPs, whereas one CP seems to be a too simple assumption over 20 years of economic evolution. Given that the CP analysis results agree with each other, we find consistent and reasonable results from sub-periods up to the full period—the retrospective analysis of the whole dataset suggests a connection between the CP PDFs of linear trend changes in the mean market correlation and major economic events as the dot-com bubble, the global financial crisis and the Euro crisis. Additionally, lower peaks seem to accumulate in crisis periods of lower impact. Of course, the analysis is not feasible to infer on causal dependencies between trend changes of the mean market correlation and single economic events, but our studies support the idea of the mean market correlation (i.e., the market mode) to be an informative measure of macroeconomic changes as stated in the literature (cf. [1,2,3]). We note that the onset of the global financial crisis is especially reflected by the CP PDF peaks of the U.S. housing bubble burst and might illustrate the strong interdependence of the bubble and the global crisis.

Furthermore, the CP PDFs divide the considered period into intervals that match rather well with the locally stable market state intervals found in an article of Stepanov et al. [1] by a clustering approach: The authors in [1] find the first four years to be a rather calm period in cluster states one and two which is mixed with some intermediate state five around Spring 1996. This is reflected by weakly pronounced CP PDF peaks at that time. A more notably transition occurs over the dot-com bubble from 1999 to 2003 into a fully intermediate transition regime dominated by states three, four, five and six. The transition is marked by the CP PDFs of the dot-com bubble and by an increased CP probability in the whole intermediate period up to 2007. Around this date the authors in Stepanov et al. define states seven and eight to be economic turbulent crisis states which is in accordance with the high CP PDFs in the end of 2006 and beginning of 2007 as well as the pronounced local CP PDF modes in the turbulent period from 2007 to the end of 2012. Interestingly, we can reconstruct approximately the economic state structure presented in Stepanov et al. by inferring them directly from the trend changes of the mean market correlation. Our independent methodological CP approach adds new evidence to the general reasoning of locally (meta)stable market states. The existence of quasi-stationary states is further supported by an independent resilience analysis on the *S&P500* mean market correlation time series which suggests locally stable economic states eventually operating on multiple time scales [12].

The online approach, assuming one CP on the pre-crisis segments that are updated with in-crisis data over time, yields further insights into the methodology’s sensitivity and the relation of CP PDFs depending on the considered time period. First, the CP PDF is consistently shifted from left to right in times of crises’ onsets for the dot-com bubble and the financial crisis. The shift becomes manifest in our analysis by including data of roughly 80 to 100 trading days after the crisis onset. The character of the Euro crisis seems to be different from the former two because of the almost direct transition from the financial crisis, i.e., missing isolation of the events in the time dimension. The CP PDF is jumping roughly between a stock market crash on 1 December 2008 and April 2009 depending on the time series length. This might indicate a cascade of non-isolated events in that period. Note that this is also reflected by the wider modes of the CP PDFs in the retrospective analysis in Section 3.1 between 2008 and 2010. Furthermore, zooming into the pre-crisis segments instead of analysing the whole $\bar{C}\left(t\right)$ time series changes the resolution of the detected trends in relation to the events included in the segments. For example, the highest peaks in the retrospective analysis are around the financial crisis onset and after the Euro crisis’ culmination event of Greece’s downgrading. More local events as the bond crisis 1994, the Mexican peso crisis or Asian and Russian financial crisis are not or less pronounced in relation to the included major events. After exclusion of the major crisis events by zooming, e.g., into the pre-dot-com bubble segment, the CP PDF accumulates around the bond crisis 1994 and Mexican peso crisis. This means the CP PDFs might reflect, up to a certain degree, an economic impact ranking of the events relative to the most important ones that are included in the analysis segment. Moreover, such an observation implies that the zoom changes the event impact resolution of the CP analysis.

Of course, the online approach cannot be seen as a precursor tool for economic crisis, since it needs a certain minimum amount of in-crisis data and more important, a recently detected CP might be identified as part of fluctuations if again new data are available. However, the overall results imply that the mean market correlation $\bar{C}\left(t\right)$ contains information about changing market dynamics and global crisis events. This gives rise to the idea of designing leading indicators and precursors based on the market mean correlation $\bar{C}\left(t\right)$. Related scientific attempts of precursor design based on correlations in general can be already found in the literature [8,9,66].

Although the presented study suggests that the mean market correlation is a promising economic indicator, it is far too early for final conclusions on the mean market correlation’s role as an economic information measure. Future research on this topic could consider other time periods and/or markets and might apply similar methodologies as ours. Additionally, studies on alternative methods might allow even for causal inference on the relation of mean market correlation and macroeconomic events.

Furthermore, the CP analysis almost always detects a CP by design, unless it is used on an unperturbed and perfectly straight line. In that sense, it might be advisable to assure our trust in the detected trend changes by involving further analyses. For example, the computed segment fit can be compared to a straight line via a Bayesian model comparison [13,17] or information criteria [67] to infer whether the number of assumed CPs is a justified condition or not. However, due to their definition, the information criteria might always prefer the straight lines for time series that include only a few data beyond the CP. Assigning a higher weight to the most recent data might be a possible way to remedy this problem.

## Figures and Tables

**Figure 1 entropy-25-01265-f001:**
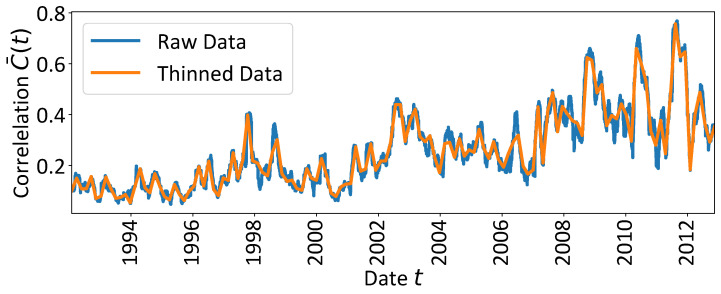
The calculated mean correlation of the *S&P500* market. The time series are the mean values of correlation matrices which were calculated based on moving windows of length τ=42 days plotted against the 21st time stamp of the τ-days-interval. The original raw correlation time series is computed on a sequence of data windows that are successively shifted by one trading day. The thinned version, which contains every 40th data point of the raw time series, i.e., 132 data points in total, is used to avoid exorbitant computation times due to the combinatorics of change point (CP) configurations involved in the CP analysis.

**Figure 2 entropy-25-01265-f002:**
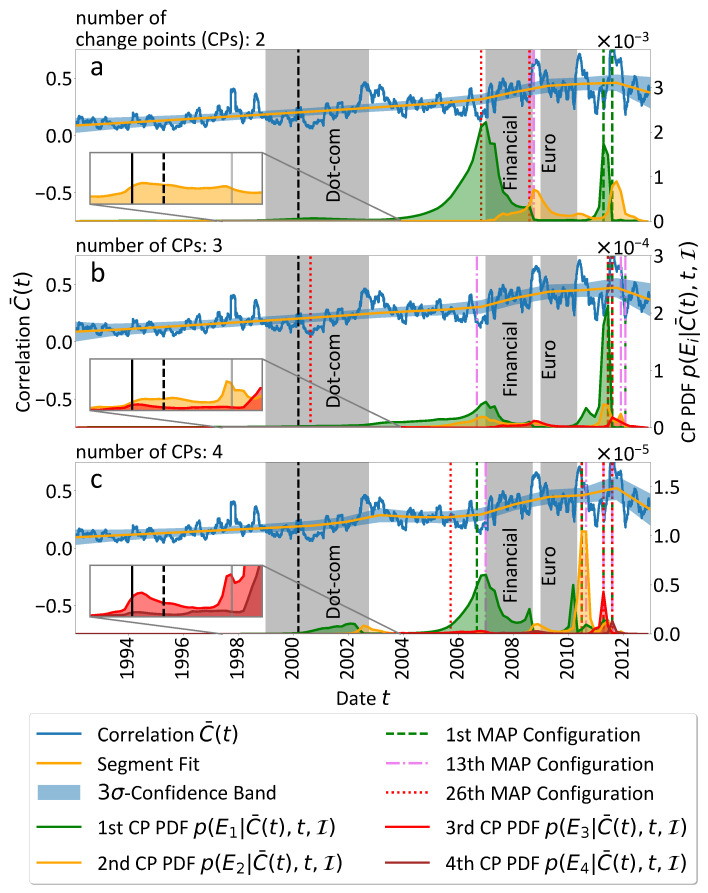
The CP analysis results under the assumption of two, three and four CPs is shown in (**a**–**c**), respectively. The *x*-axis ticks correspond to the 1 January of each year. The major economic events, such as the dot-com bubble, the global financial crisis, and the Euro sovereign debt crisis, are indicated by the grey shaded areas. The dot-com bubble’s rise starts around 1 January 1999, reaches its all time high at the dashed vertical line, and ends around 3 October 2022 with an after-bubble trough. The financial crisis starts with the burst of the U.S. housing bubble around 1 January 2007 and culminates with the Lehman Brothers’ bankruptcy on 15 September 2008, which marks the end of the gray shaded area. The Euro crisis starts with the first insolvencies of European banks, starting with 1 January 2009 and culminates in the downgrading of Greece’s sovereign debt rating on 27 April 2010. The shown CP probability density functions (PDFs) exhibit consistent peaks in the areas of major economic disturbance. Furthermore, three of the most probable CP configurations are indicated by vertical lines as an example. They are in good agreement with the CP PDFs and the signed crisis events. The insets zoom into some peaks of the higher order CP PDFs to illustrate that their profile also contains valuable information, e.g., for the dot-com bubble in (**a**,**b**). At least the first CP PDF in (**a**) is slightly increased shortly after the dot-com bubble’s all time high, but in (**b**) only the inset provides insights into the dot-com bubble period. The linear segment fit with corresponding confidence intervals is the PDF-weighted average of the linear segment fits at that time and is shown in orange to illustrate the agreement between (**a**–**c**). For a detailed discussion of the results we refer to the running text. Additional results for the assumption of one and five CPs can be found in Appendix B.

**Figure 3 entropy-25-01265-f003:**
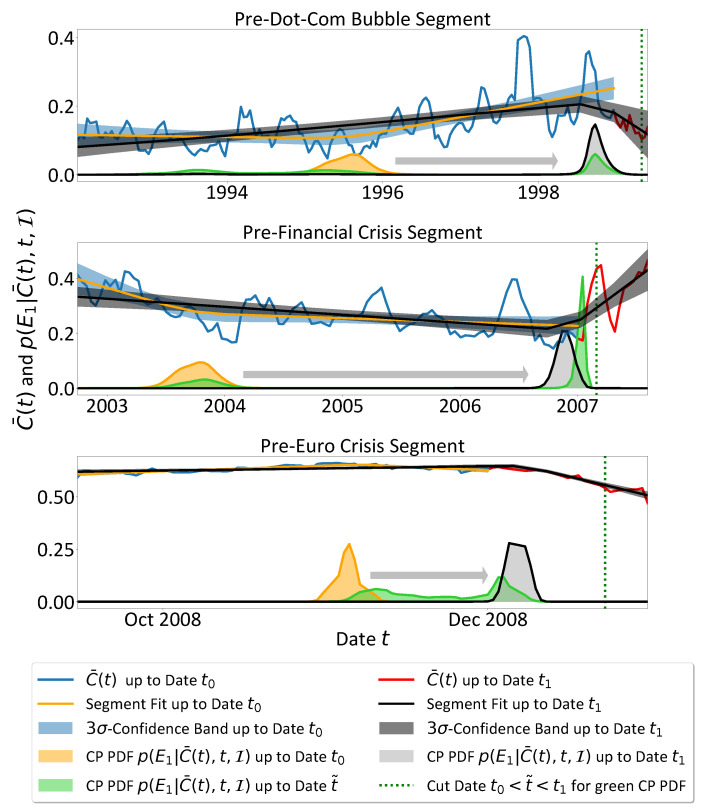
Results of a CP analysis on specific subsets of the whole mean market correlation $\bar{C}\left(t\right)$ under the assumption of one CP. The blue pre-crisis segments start on 31 January 1999 (time series onset), 1 October 2002 (end dot-com bubble) and 15 September 2008 (Lehman Brothers’ bankruptcy) for the dot-com bubble, the financial crisis and the Euro crisis, respectively. They end with the dates $t_{0}$ at the crisis onsets on 1 January 1999 (dot-com bubble), 1 January 2007 (financial crisis), and slightly before the crisis onset on 1 December 2008 (Euro crisis). The time series are thinned by using every tenth data point for the dot-com bubble and the financial crisis. The resulting orange CP PDFs are compared to the black PDFs that are computed on the red updated time series segments up to dates $t_{1}$, i.e., 1 June 1999 (dot-com bubble), 9 August 2007 (financial crisis) and 1 January 2009 (Euro crisis). As expected the major crisis onsets are captured by the black CP PDFs after some in-crisis data are included, whereas the orange CP PDFs may reflect the consequences of economic events confined to the segment interval. Interpreted from an online analysis perspective, the appearance of a pronounced and narrow peak (cf. black CP PDFs) might help to identify important current economic events that may influence future economic states more than the ones detected by the orange smaller an wider CP PDFs. In contrast to the more pronounced black CP PDF in the former two cases, a higher similarity of the orange and black CP PDF is observed in the Euro crisis case. This might indicate the difficulties of isolating the Euro crisis from the financial crisis as a cascade of events implies lots of trend changes with an overall negative mean market correlation trend in the end of 2008. The green interim CP PDFs are computed on data up to the green dotted vertical lines at date $\tilde{t}$ and mark the minimum amount of updated data that is needed to locate the global maximum of the green CP PDF less than 100 trading days apart from the defined crisis onset dates. The segment fits with corresponding confidence bands (CBs) are shown to guide the eye. More information can be found in the running text.

## Data Availability

The prepared data for the change point analysis, the analysis results and corresponding Python codes are available on Zenodo via https://doi.org/10.5281/zenodo.8186951 (accessed on 23 August 2023) under a *Creative Commons Attribution 4.0 International*. The open source python-implementation is named *antiCPy* and can be found at https://github.com/MartinHessler/antiCPy (accessed on 23 August 2023) under a *GNU General Public License v3.0*. The raw data can be downloaded via [68] and the preprocessed time series is also available via [69].

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
