# Peer review of "Efficient Multi-Change Point Analysis to Decode Economic Crisis Information from the S&P500 Mean Market Correlation"

_entropy, 2023, doi:10.3390/e25091265_

Round 1

Reviewer 1 Report

This paper presents a Bayesian multi-trend change point analysis method implemented in Python to identify significant macroeconomic events responsible for dramatic changes in the economy. The analysis focuses on the mean market correlation of the S&P500 stock time series over roughly 20 years, including the dot-com bubble, global financial crisis, and Euro crisis. The authors demonstrate the method in two ways: first, retrospectively on the entire dataset, and second, in an online adaptive manner on pre-crisis segments.

The paper seems to show that the mean market correlation is a promising economic indicator, but further research is needed to establish its role as an informative measure for macroeconomic events. 

My main comments are reported below:

1) Please proofread the written English, especially in the Abstract, Introduction, and Conclusions sections, as the current text is challenging to read (you can use Grammarly.com, chatGPT, etc).

2) Unfortunately, it is unclear what the economic utility of the mean market correlation is, given that the methodology is not new. It would be helpful to propose some practical economic and financial applications.

3) This approach completely disregards the fact that financial market returns do not follow a normal distribution and are highly asymmetric and leptokurtic.

4) The quality of the figures is significantly low and needs substantial enlargement as they are difficult to read in their current state.

Please proofread the written English, especially in the Abstract, Introduction, and Conclusions sections, as the current text is challenging to read (you can use Grammarly.com, chatGPT, etc).

Author Response

Dear Anonymous Reviewer 1,

we thank you for your quick and valuable feedback upon our manuscript. It definitely improved the manuscript's quality. We discussed all comments and questions in the attached answer letter. The revised manuscript as well as the version with changes highlighted will be uploaded separately as I understand the upload mask.

Sincerely yours,

Martin Heßler on behalf of all authors

Reviewer 2 Report

Dear Authors,

Thank you for the opportunity to read and review your paper. I appreciate your research idea and how did you use the econometric models based on time series analysis. The models and research approach are correctly used and also the research results are relevant for the special issue in Entropy. Also, I appreciate that you have used relevant sources from the literature in order to base the research idea and you have integrated correctly you research results in the current research trend.

I encourage you to continue research in the field.

Best regards,

Anonymous Reviewer

Author Response

Dear Anonymous Reviewer 2,

We thank you for your quick response and are glad to hear that you appreciated reading the manuscript. Please find attached our answers to all reviewers' comments and questions.

Sincerely yours,

Martin Heßler on behalf of all authors

Round 2

Reviewer 1 Report

.

.